# Timers and Such: A Practical Benchmark for Spoken Language Understanding with Numbers

**Loren Lugosch**
McGill University / Mila
lugoschl@mila.quebec

**Piyush Papreja**
ppapreja@asu.edu

**Mirco Ravanelli**
Université de Montréal / Mila
mirco.ravanelli@gmail.com

**Abdelwahab Heba**
Université Paul Sabatier, IRIT, CNRS
aheba@irit.fr

**Titouan Parcollet**
Avignon Université
titouan.parcollet@univ-avignon.fr

## Abstract

This paper introduces Timers and Such, a new open source dataset of spoken English commands for common voice control use cases involving numbers. We describe the gap in existing spoken language understanding datasets that Timers and Such fills, the design and creation of the dataset, and experiments with a number of ASR-based and end-to-end baseline models, the code for which has been made available as part of the SpeechBrain toolkit.

## 1 Introduction

Spoken language understanding (SLU) research has begun to emphasize the importance of both *testing* and *training* SLU systems end-to-end on audio. *Testing* on audio is important because an independently trained automatic speech recognition (ASR) system and natural language understanding (NLU) system will not necessarily work well when combined [1, 2]. *Training* SLU systems end-to-end on audio is likewise worthwhile because it can make the NLU model more robust to transcription errors, and because it enables training a single neural network to perform the entire SLU pipeline without an intermediate search step, a technique with many practical and theoretical advantages over ASR-based approaches [3].

Experiments involving end-to-end training and testing of SLU models require audio data. Over the last few years, a number of open source audio datasets have been released to enable high-quality, reproducible end-to-end SLU research. The **Snips SLU Dataset** [2] is a small dataset of English and French commands for a smart home setting, such as controlling smart lights, speaker volume, and music selection. **Fluent Speech Commands** [4] is a somewhat larger, though simpler, dataset of similar English smart home commands. The most recently released **SLURP** dataset [5] is an even larger and much more semantically complex multi-domain SLU dataset.

An important feature missing from these datasets is a thorough coverage of *numbers*. Numbers are necessary for many SLU domains, especially for very common use cases like setting timers and converting units of measurement while cooking. While there do exist datasets of digits spoken in isolation [6, 7, 8], and the Snips SLU Dataset and SLURP do have a small number of commands involving simple numbers, there does not to our knowledge exist any open source SLU dataset that covers more general multi-digit numbers (e.g. "13.57", "-21.4") spoken in context. The dataset

35th Conference on Neural Information Processing Systems (NeurIPS 2021) Track on Datasets and Benchmarks.

```
1  ("what's 37.67 minus 75.7",
2  {
3    'intent': 'SimpleMath',
4    'slots': {
5      'number1': 37.67,
6      'number2': 75.7,
7      'op': ' minus '
8    }
9  })
```

Listing 1: A `SimpleMath` command and its label dictionary.

introduced here—**Timers and Such**—fills this gap, with each command containing one or two numbers with one or more digits.

One of the original motivations for the development of end-to-end SLU models was the need for more compact models that can easily fit on resource-limited devices and operate without an Internet connection [3]. Whereas existing SLU datasets focus mostly on Internet-connected smart home commands or queries that require an Internet search, Timers and Such is composed only of commands that can be executed without the need for the Internet. This makes the dataset ideal for training or testing a simple offline voice assistant. While the baselines described in this paper all use rather comfortably large neural networks ($>$100 million parameters), we hope that researchers and developers working on machine learning for edge devices will improve upon our models in terms of storage requirements and computational complexity; we believe they will find Timers and Such to be a challenging and interesting test case for their models.

The dataset should also be useful for researchers working on representation learning for audio and language to use as a downstream test task, as Fluent Speech Commands has been [9, 10]. While in the past we have found supervised ASR-based pre-training to be essential for getting good results with end-to-end SLU models, we believe unsupervised feature extractors may ultimately prove to be a better general-purpose solution for SLU and other audio tasks [11, 12].

A final, more mundane motivation for Timers and Such was the need for an SLU dataset that could easily be downloaded programmatically using tools like `wget` or `curl`, similar to MNIST or LibriSpeech.[1] Fluent Speech Commands requires users to sign up on a web page, and the Snips SLU dataset requires filling in an online form and waiting to be approved. In contrast to these, Timers and Such is hosted on Zenodo[2] under the very permissive CC0 license, and the experiment code[3] we provide downloads the dataset if it is not already present in the location specified by the user. These features should lower the barrier to entry for anyone interested in training or testing their first SLU model.

In what follows, we outline the design and creation of Timers and Such, describe some baseline models for the dataset, discuss their experimental performance, and end by listing some ideas for future research.

## 2  Dataset design

The dataset has four intents, corresponding to four common offline voice assistant uses: `SetTimer`, `SetAlarm`, `SimpleMath`, and `UnitConversion`. The semantic label for each utterance is a dictionary with the intent and a number of slots. An example of a command and its corresponding semantics is shown in Listing 1.

The prompts to be recorded by speakers were generated using a script written by the first author with a simple "grammar" that produced a few variations of set phrases for each of the four intents ("set a timer for...", "set timer for...", "start timer for..."). Random numbers were inserted from a range

---

[1] SLURP, released after the start of this work, can also be downloaded programmatically.

[2] The dataset can be found at https://zenodo.org/record/4623772.

[3] The code can be found at https://github.com/speechbrain/speechbrain/tree/develop/recipes/timers-and-such.

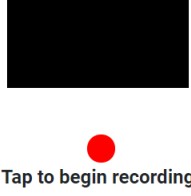

Figure 1: The recording interface used by speakers.

that made sense for the given intent (for instance, when converting temperatures, temperatures less than 0 Kelvin were not used).[4]

A better way to collect different ways of phrasing commands than introspection is to place speakers in a voice control scenario (or have them imagine themselves in one) and ask them what they would say to have the system complete a certain task. This method was used to create part of the closed source Facebook dataset in [3] and the open source SLURP [5]. However, this approach is complicated to set up and much more taxing on speakers. Given that our speakers were volunteers, we decided instead to simply prompt them with randomly generated phrases for each of the intents, similar to the approach used in Mozilla's Common Voice project [13].

## 3 Preliminary small-scale study

A preliminary version of Timers and Such was made between November 2019 and October 2020. 11 colleagues recorded themselves reading a list of prompts, some using the first author's laptop, and others using their own computers. The first author then segmented these audio files into the individual commands and split the resulting 271 audios into a training set with 144 audios (4 speakers), a dev set with 72 audios (2 speakers), and a test set with 55 audios (5 speakers). Models trained on this small dataset were found to have high variability in performance for the test set, which was hypothesized to be because of the small test set size. (This actually seems not to have been the real reason; see Sec. 5.4.) To make a dataset that could be used more reliably to train and compare SLU models, we decided to reach out to a larger pool of speakers by asking volunteers online to donate their voices.

## 4 Data collection

### 4.1 Recording website

The second author built a website to allow speakers to record themselves reading prompts. Speakers using the website were first asked for their age, gender, and spoken English proficiency. For each demographic field, users also had the option to respond "Prefer not to say". After giving their consent to have their demographic information and recordings released in a publicly available dataset, speakers used the interface shown in Fig. 1 to record a set of 24 randomly generated prompts.

---

[4]The script for generating prompts can be found at https://gist.github.com/lorenlugosch/5df9e30227aa5c67ff51cd28271414f0.

## 4.2 Speaker recruitment

Starting on February 18, 2021, we advertised the project and recording website on various social media platforms (Twitter, LinkedIn, Reddit, Hacker News, Facebook). In response to this advertisement, 89 sessions were recorded from the first day until March 12, 2021.

Whether the 89 recorded sessions correspond to exactly 89 different speakers is unknown. We neglected to ask speakers in the recording instructions not to record more than one session. Because speakers were (deliberately) not asked to provide any information that would uniquely identify them, such as their name or email address, there is no way to ascertain whether two sessions correspond to the same speaker (as is the case for recording platforms like Common Voice's, which allow a speaker to record without entering any personally identifiable information). To avoid an overlap between speakers in the training set and the test set, we examined the demographic information provided by speakers (age, gender, fluency) and selected only sessions with a unique demographic triple to be in the test set. Assuming speakers provided their demographic information truthfully, this means there are no speakers from the test set in the training set.

## 4.3 Data preprocessing and cleaning

All recordings were converted from their original formats to single-channel 16,000 Hz .wav files for compatibility with the ASR model used in our baseline experiments.

Data cleaning for the smaller set of audios collected during the preliminary small-scale study was done manually by the first author. The 271 audios collected in the preliminary study were assigned to the dev-real subset. Those speakers were not asked for their demographic information, so that information is not provided for this split.

For the larger set of audios recorded using the recording website, we used a more automated form of cleaning: the audios were transcribed using an ASR model (described in Sec. 5.1), and the word error rate (WER) between each prompt and transcript was computed. Audios for which the ASR transcript was empty or looked significantly different from the prompt were listened to and kept or deleted as appropriate. (A simple automatic decision rule that was found to yield nearly the same subset was to select all audios with WER less than 100%.) After this cleaning procedure, the remaining 1,880 audios were split into train-real and test-real subsets. A .csv file for each subset ({train-real, dev-real, test-real}.csv) lists, for each utterance, the .wav filename, the semantic label dictionary, the session ID ($\approx$ speaker ID), and the transcript.

## 4.4 Synthetic data

Following [14], we used VoiceLoop [15] to synthesize a large set of audios from 22 synthetic speakers. (The VoiceLoop model is trained on the VCTK dataset [16].) That set was split by speaker into the train-synth (192,000 audios), dev-synth (24,000 audios), and test-synth (36,000 audios) subsets. As for the data from the real speakers, we include a .csv file ({train-synth, dev-synth, test-synth}.csv) listing the filename, semantics, speaker ID (a number 1 to 22 indicating which VoiceLoop synthetic speaker was used), and transcript. The VoiceLoop speech synthesizer is deterministic: running it on the same prompt twice produces the same audio signal. As a result, some of the rows in the .csv file describing the synthetic subset are redundant: they point to the same audio file with the same labels. We have not removed the redundant rows because we found that doing so led to an unbalanced training set: for example, there were many more instances of "set alarm for <hour> <minute> AM" than of "set alarm for <hour> AM", so models trained on this unbalanced dataset tended to hallucinate an erroneous value for the <minute> slot for the latter type of utterance. (Alternately, users can rebalance the data in a different way, if they choose, using e.g. pandas.DataFrame.drop_duplicates() on the filename column of the .csv file.) We encourage users of Timers and Such not to think of the synthetic subset as *fixed* (except to avoid unfair comparisons between two models differing in some other respects), but rather to try adding more synthetic speakers and using improved speech synthesis techniques.

## 4.5 Dataset statistics

The overall statistics for both the real and synthetic subsets of Timers and Such after data cleaning are listed in Table 1. At 2,151 non-synthetic utterances, Timers and Such is a fairly small dataset,

but like TIMIT (6,300 utterances [17]) and the Snips "smart lights" dataset (1,660 utterances [2]), we have found the dataset nonetheless very useful for experimentation. It is more challenging than Fluent Speech Commands (which can be treated as a simple classification problem and for which accuracy as high as 99.7% has been achieved [18]), but it is smaller and simpler than SLURP. By training only on text or synthetic speech, and testing on all available real audio, it is possible to obtain a relatively large test set (cf. the LibriSpeech test-clean subset with 2,620 audios).

Table 1: Timers and Such speaker counts and recording statistics. (*Speaker counts are approximate; see Section 4.2.)

| Split | # of speakers* | # of audios | # hours |
|---|---|---|---|
| train-synth | 16 | 192,000 | 132.2 |
| dev-synth | 2 | 24,000 | 15.8 |
| test-synth | 3 | 36,000 | 23.5 |
| train-real | 74 | 1,640 | 1.9 |
| dev-real | 11 | 271 | 0.3 |
| test-real | 10 | 240 | 0.3 |
| all-real | 95 | 2,151 | 2.5 |

Table 2: Speaker gender statistics. (dev-real demographics not included; see Section 4.3.)

| Split | Man | Woman | Non-Binary | (Prefer not to say) |
|---|---|---|---|---|
| train-real | 54 | 17 | 0 | 3 |
| test-real | 5 | 4 | 1 | 0 |

Table 3: Speaker English proficiency statistics.

| Split | Native speaker | Fluent | Somewhat fluent | (Prefer not to say) |
|---|---|---|---|---|
| train-real | 20 | 42 | 9 | 3 |
| test-real | 4 | 2 | 4 | 0 |

# 5  Baseline models

Here we describe extensive experiments with a set of baseline neural network models for Timers and Such. All experiments are conducted using the open source SpeechBrain [19] toolkit.

## 5.1  ASR model and language models

The baseline models use an ASR model trained on the 960-hour LibriSpeech English ASR dataset [20]. The ASR model is an autoregressive attention-based sequence-to-sequence model [21, 22] that achieves 3.08% WER on the test-clean subset of LibriSpeech. The encoder of the ASR model extracts 40-dimensional FBANK features from the input signal and has two 2-D convolutional layers that downsample the input sequence by a factor of 4 in the time dimension, followed by four bidirectional LSTM layers and two fully-connected layers. The decoder is a GRU network that uses the location-aware attention mechanism of [23] to process the encoder outputs. The encoder outputs are additionally passed through a linear CTC [24] head; during training, the output of the CTC head is used to compute an auxiliary CTC loss term [25]. Both the CTC head and the autoregressive decoder have 1000 outputs for a 1000-token SentencePiece [26] BPE vocabulary.[5] (This ASR model was

---

[5]More detailed hyperparameters for the ASR model can be found at `https://github.com/speechbrain/speechbrain/blob/develop/recipes/LibriSpeech/ASR/seq2seq/hparams/train_BPE_1000.yaml`.

Table 4: Speaker age ranges. (See `train-demographics.csv` and `test-demographics.csv` for more granularity.)

| Split | 18-25 | 26-35 | 36-45 | 46+ | (Prefer not to say) |
|---|---|---|---|---|---|
| train-real | 11 | 41 | 6 | 1 | 15 |
| test-real | 3 | 5 | 2 | 0 | 0 |

chosen because it was the best performing English ASR model in SpeechBrain at the time when these experiments were conducted.)

The ASR model transcribes the input signal $\mathbf{x}$ using a beam search for

$$\underset{\mathbf{y}}{\operatorname{argmax}} \quad \log p_{\text{ASR}}(\mathbf{y}|\mathbf{x})$$
$$+ \alpha \log p_{\text{CTC}}(\mathbf{y}|\mathbf{x})$$
$$+ \beta \log p_{\text{LM}}(\mathbf{y})$$
$$+ \gamma c(\mathbf{x}, \mathbf{y}),$$

where $p_{\text{CTC}}(\mathbf{y}|\mathbf{x})$ is the likelihood of transcript $\mathbf{y}$ according to the CTC head [25], $p_{\text{LM}}(\mathbf{y})$ is the likelihood according to an external language model (LM), $c(\mathbf{x}, \mathbf{y})$ is a coverage penalty term [27], and $\alpha, \beta, \gamma$ were set to minimize WER on the LibriSpeech dev sets.

The default LM is an LSTM trained on the LibriSpeech language modeling resources.[6] In addition to the default LibriSpeech LM (LS LM), we also trained an LSTM LM on the Timers and Such training set transcripts (TAS LM). For ASR-based baseline models, we present results both using the LS LM and TAS LM.

## 5.2 SLU models

We provide code, pre-trained models, and results for a traditional decoupled SLU model and (using the terminology suggested by Haghani et al. in [28]) two types of "end-to-end" models: a multistage model and a direct model.

The **decoupled** model uses a sequence-to-sequence model to map the transcript to the semantics. During training (and when decoding the validation set), the ground-truth transcripts are used as the input, and during testing, the transcripts produced by the LibriSpeech ASR model are used. For all models, the semantic dictionaries are treated as raw sequences of characters and split using a 51-token SentencePiece tokenizer.

The **multistage** model likewise uses a sequence-to-sequence model to map the transcript to the semantics, but instead of training on the ground-truth transcripts, it is trained on the ASR transcripts. The transcripts are not precomputed: rather, each minibatch of audio signals is transcribed on the fly during training, which simplifies the implementation of our experiments. In theory, transcribing training examples on the fly should also make the NLU model more robust, as it is exposed to more types of transcription errors resulting from different noise samples (e.g. from dropout, batch normalization, data augmentation) across minibatches—though we have not compared the results with simply training on a single set of precomputed ASR transcripts, and leave this as an avenue for other researchers to explore. The downside of on-the-fly transcription is that the inherently sequential ASR beam search becomes a bottleneck on training step time. Using the default ASR beam width of 80, the time for one epoch on `train-synth` was about 12 hours (compared with about 0.5 hours for the decoupled model). Reducing the ASR beam width to 1 reduced the time for one epoch to about 2.5 hours. The results presented below use an ASR beam width of 1 for the multistage model.

The **direct** model uses a single sequence-to-sequence model to map audio directly to semantics, without an intermediate ASR search step. Compared to the multistage model, the direct model is significantly faster both in training and decoding, at about 1.5 hours per epoch with `train-synth` instead of 2.5 hours. Pre-training using related ASR or NLU tasks has consistently been found to improve the performance of direct models [4, 29, 30, 31, 32], so we pre-train the encoder here as well.

---

[6]https://www.openslr.org/11/

In our experiments described in previous papers, the encoder of the direct model was pre-trained using force-aligned phoneme and word labels [4, 14]. The pre-training strategy used in this paper is somewhat simpler: we extract the encoder from the LibriSpeech ASR model and use it as a feature extractor in the direct SLU model. Another difference is that we do not backpropagate into the pre-trained encoder and leave its weights frozen, which greatly reduces training time and memory consumption. A more thorough ablation study and comparison of pre-training strategies would be worthwhile to conduct, but we leave that for the future, since the point here is just to establish some reasonable baseline models for this dataset.

While the SLU models do use a beam search to produce the output sequence, there are a number of differences between the SLU decoder and the ASR decoder. The SLU beam search does not use a coverage penalty (which was found to hurt performance both for Timers and Such and for the SLURP dataset) or an external "language model" over the space of output dictionaries. Instead of location-aware attention (which assumes a monotonic alignment between input and output sequences), the SLU decoder uses a simple one-headed key-value attention mechanism. The SLU models also do not use an auxiliary CTC head: whereas CTC's assumptions (monotonic alignments; output length < input length) make sense for ASR, they generally do not hold for SLU, unless the dataset has word-aligned slot labels (Timers and Such does not). Other hyperparameters for these models were not optimized and chosen simply by copying the decoder hyperparameters from the LibriSpeech recipe, which were optimized for the validation set of that dataset.

### 5.3  Experiments

For all baseline models, we provide results for three composite training sets: `train-real` only (trained for 50 epochs), `train-real` plus `train-synth` (trained for 2 epochs), and `train-synth` only (trained for 2 epochs). For all three training sets, we measure performance on `test-real` and `test-synth`. When training on `train-synth` only, we additionally report performance for `all-real`, a subset obtained by combining all the real data in `train-real`, `dev-real`, and `test-real`. (We do not test models trained on `train-real` on `all-real` because `all-real` contains `train-real`. For the same reason, we use `dev-synth`, not `dev-real`, to select the model checkpoint from the epoch with the best validation performance when testing on `all-real`.)

As in previous work, we report performance in terms of accuracy, where an output is deemed "correct" if all predicted slots and slot values are correct. Bastianelli et al. in [5] have argued for the use of metrics more informative than simple accuracy when evaluating end-to-end SLU models. They propose SLU-F1, a metric based on word-level and character-level edit distance between the model's output and the true labels. The SLU-F1 metric sensibly penalizes errors like "pizzas" → "pizza" less than errors like "pizzas" → "fries". It is unclear, though, whether character-level edit distance is suitable for the numeric commands of Timers and Such: should "11" → "111" (character error rate of 50%) be regarded as less of an error than "11" → "22" (character error rate of 100%) when setting a cooking timer in minutes? For this reason, we do not recommend using character-level error to evaluate systems for this task. As a compromise, we also suggest reporting "SLU WER", an easy-to-compute metric that treats the space-delimited output of the SLU model and the true output dictionary as regular sequences of words and simply computes the usual WER metric. Note that no "normalization" of the outputs (e.g., "twelve and a half", "twelve point five" → "12.5") is necessary before evaluating, since the labels are always written in the correct numeric format.

### 5.4  Results

A few trends in the results shown in Table 5 are worth noting.

- **The direct model and multistage TAS LM work best.** This is perhaps unsurprising, since these two models effectively have the most opportunity to train on the downstream SLU task.

- **The direct model "overfits" to synthetic speech.** It seems that because the direct model has access to the raw speech features instead of a transcript, it can learn the idiosyncratic pronunciations of the speech synthesizer and achieve much better performance than the ASR-based models (96.7% vs. 85.4%). This model still performs well on the real test data—we mention this simply to explain why this model suddenly performs so much better for the synthetic test data.

Table 5: Results (mean and stdev. over 5 random seeds) for all baseline models. See Sec. 5.3 for the definition of "SLU WER".

| Model | Training set | test-real | | test-synth | |
|---|---|---|---|---|---|
| | | Accuracy | SLU WER | Accuracy | SLU WER |
| Decoupled (LS LM) | train-real | $24.1\%_{\pm1.1\%}$ | $34.4\%_{\pm3.3\%}$ | $16.1\%_{\pm1.4\%}$ | $33.2\%_{\pm8.7\%}$ |
| | (both) | $31.4\%_{\pm4.3\%}$ | $26.5\%_{\pm5.0\%}$ | $22.5\%_{\pm2.1\%}$ | $25.2\%_{\pm2.5\%}$ |
| | train-synth | $32.3\%_{\pm3.9\%}$ | $26.5\%_{\pm2.5\%}$ | $23.7\%_{\pm1.6\%}$ | $24.2\%_{\pm0.7\%}$ |
| Decoupled (TAS LM) | train-real | $43.5\%_{\pm2.0\%}$ | $20.3\%_{\pm3.5\%}$ | $34.6\%_{\pm1.2\%}$ | $18.5\%_{\pm3.8\%}$ |
| | (both) | $46.8\%_{\pm2.1\%}$ | $16.5\%_{\pm2.2\%}$ | $38.4\%_{\pm1.3\%}$ | $15.2\%_{\pm0.9\%}$ |
| | train-synth | $49.1\%_{\pm2.3\%}$ | $16.3\%_{\pm1.1\%}$ | $39.9\%_{\pm0.7\%}$ | $13.9\%_{\pm0.8\%}$ |
| Multistage (LS LM) | train-real | $55.5\%_{\pm3.4\%}$ | $10.1\%_{\pm0.6\%}$ | $43.1\%_{\pm2.9\%}$ | $10.8\%_{\pm0.8\%}$ |
| | (both) | $67.8\%_{\pm1.4\%}$ | $7.4\%_{\pm0.4\%}$ | $79.4\%_{\pm0.4\%}$ | $3.2\%_{\pm0.1\%}$ |
| | train-synth | $66.6\%_{\pm0.8\%}$ | $7.7\%_{\pm0.8\%}$ | $79.1\%_{\pm0.2\%}$ | $3.2\%_{\pm0.0\%}$ |
| Multistage (TAS LM) | train-real | $64.0\%_{\pm3.3\%}$ | $7.4\%_{\pm0.9\%}$ | $51.5\%_{\pm2.9\%}$ | $8.7\%_{\pm0.7\%}$ |
| | (both) | $72.6\%_{\pm1.6\%}$ | $5.9\%_{\pm0.1\%}$ | $85.4\%_{\pm0.2\%}$ | $2.4\%_{\pm0.0\%}$ |
| | train-synth | $72.2\%_{\pm1.4\%}$ | $6.2\%_{\pm0.4\%}$ | $85.4\%_{\pm0.3\%}$ | $2.4\%_{\pm0.1\%}$ |
| Direct | train-real | $\mathbf{81.6\%}_{\pm5.4\%}$ | $\mathbf{2.6\%}_{\pm1.1\%}$ | $70.0\%_{\pm5.7\%}$ | $15.2\%_{\pm19.1\%}$ |
| | (both) | $77.5\%_{\pm1.6\%}$ | $3.3\%_{\pm0.4\%}$ | $\mathbf{96.7\%}_{\pm0.3\%}$ | $\mathbf{1.1\%}_{\pm0.0\%}$ |
| | train-synth | $68.0\%_{\pm5.5\%}$ | $8.9\%_{\pm3.4\%}$ | $96.4\%_{\pm0.2\%}$ | $\mathbf{1.1\%}_{\pm0.0\%}$ |

Table 6: Baseline results for the all-real set.

| Model | Training set | all-real | |
|---|---|---|---|
| | | Accuracy | SLU WER |
| Decoupled (LS LM) | train-synth | $26.8\%_{\pm3.3\%}$ | $29.0\%_{\pm2.2\%}$ |
| Decoupled (TAS LM) | train-synth | $44.6\%_{\pm2.4\%}$ | $17.3\%_{\pm1.1\%}$ |
| Multistage (LS LM) | train-synth | $64.6\%_{\pm0.7\%}$ | $7.2\%_{\pm0.2\%}$ |
| Multistage (TAS LM) | train-synth | $\mathbf{69.9\%}_{\pm0.9\%}$ | $\mathbf{6.0\%}_{\pm0.2\%}$ |
| Direct | train-synth | $68.9\%_{\pm5.4\%}$ | $8.2\%_{\pm3.4\%}$ |

- **Test accuracies and SLU WERs[7] have high variability.** Some test accuracies have a standard deviation as high as 5.7%. We observed this phenomenon with the preliminary version of Timers and Such and suspected that the variance was because of the smaller test set size (55 audios). However, this does not seem to be the explanation here, since all-real (Table 6) has 2,151 audios and still has highly variable test accuracy (stdev. of 3.3%, 2.4%, 0.7%, 0.9%, 5.4%). We will not venture further here to diagnose this problem; instead, we leave it as a problem for future research on this dataset to solve.

## 5.5 Computing resource usage

Training and testing all the SLU models across all random seeds, models, and training set compositions required about 233 GPU-hours on an Nvidia Quadro RTX 8000 GPU. Additionally, the LibriSpeech ASR model was trained using one Nvidia Tesla V100 GPU for 194 hours, and the LibriSpeech LM was trained using 4 V100s for about 84 hours.

However, we hasten to note for those with limited computing resources interested in experimenting with Timers and Such that i) the pre-trained LibriSpeech models are available online and are downloaded automatically by the recipes, and ii) training a *single* model on Timers and Such can be done

---

[7]The 19.1% stdev. in SLU WER for the direct model on test-synth is due to a single outlier random seed for which the decoder produced many infinitely looping outputs ("unit1 unit1 unit1 unit1...").

relatively quickly, at around a minute per epoch for the direct recipe when training on `train-real`. The decoupled recipe can also be sped up significantly by using a larger batch size during training, since the input is text instead of speech and requires less memory. Note also that all the recipes have also been successfully tested on an older 12 GB Nvidia Tesla K80 GPU without any hyperparameter modifications.

## 6   Potential social impact

A risk of recording speech data is that a malicious actor could use the data to imitate the speaker and use the speaker's voice for purposes the speaker did not intend [33]. Similar to Common Voice, it is unlikely that this could happen to the speakers of Timers and Such, since they did not provide any information that could uniquely identify them.

On the whole, we think Timers and Such will be a great benefit to the research community and (indirectly) to users of voice interfaces. Speech datasets are often recorded by professional speakers in clean conditions unlike the conditions in which voice interfaces are typically used. This leads to brittle, overfitted models that break when applied to real-world speech [34]. Timers and Such will contribute to research and development of more robust models that can understand speech in a variety of accents and conditions.

## 7   Conclusion

Timers and Such is a new dataset of numeric commands that should be useful for SLU researchers, hackers aiming to train their own offline voice assistant, and researchers developing new representation learning methods for audio and language [9, 10, 11, 12] looking for another downstream task to test on. Some directions for the future of Timers and Such we hope to see worked on include: diagnosing and fixing the high variability of test performance; exploring the ASR model architecture (e.g., using a CTC model or transducer model [35]); speeding up the multistage approach, e.g. by using transfer learning to initialize a multistage model using a decoupled model; improving the performance of the direct model on `all-real`; using an ASR dataset with a more diverse set of accents and recording conditions, like Common Voice [13]; using different tokenizers or other hand-crafted output labels; improving the speech synthesis (using systems such as the RTVC multispeaker TTS [36, 37] to add even more synthetic speakers) and balance between real and synthetic training data; and enabling streaming inference [38, 39], which cannot be performed with the baseline models as-is, due to their global attention mechanism.

## Acknowledgments

Thanks to Olexa Bilaniuk for help with using the Mila cluster, Ju-Chieh Chou and Brian S. Yeh for writing the LibriSpeech recipes and beam search utilities, and Aku Rouhe and Peter Plantinga for designing and implementing many nice features of SpeechBrain that made the experiments for this paper a lot easier to run.

Timers and Such would not have been possible without the speakers who kindly took the time to donate their voices to the dataset and the friends who shared the project advertisement on social media.

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
