# OpenReview forum: "Timers and Such: A Practical Benchmark for Spoken Language Understanding with Numbers"
_NeurIPS.cc/2021/Track/Datasets_and_Benchmarks/Round1 — NeurIPS 2021 Datasets and Benchmarks Track (Round 1)_

### Official Review · Reviewer_Bag3 · 2021-06-28
**An interesting dataset requiring more insight.**

**Rating:** 6
**Confidence:** 5

**Strengths:**

-- After revision from the authors --
I thank the authors for clarifying the raised points. I believe the newer version of the paper takes care of multiple weaknesses, and thus, I lean more positively.

1. This is the first dataset to contain statements with numbers in it. The statements often also correspond to numerical operations like unit conversion, and models can be trained for such tasks.

2. There are four types of spoken commands collected in the dataset, which relate well to what users use in general.

3. The authors benchmark three techniques on their dataset. They use a decoupled technique where a pre-trained ASR is used to transcribe speech followed the NLU. They also have a multi-stage technique and a direct technique. The authors have used two metrics to report the results.

4. I like how the authors split the train & test set based on demographics and location providing nice unseen variations.

5. The dataset is public, and the codebase is pretty popular.

**Weaknesses:**

1. The dataset contains only 2.5 hours of real speech (only 1.9 hrs in the train set). The authors use synthetically generated data to augment the dataset which is 68 times more than the real data. Such less amount of real speaker data is a weakness in my view for a dataset paper.

2. The authors don't show a clear comparison table with the previous datasets for this problem. Even though they cite the relevant datasets, a comparison between the number of hours available in the previous dataset and the proposed one will be a good addition to the paper.

3. The synthetic data can be ideally collected with more variations than what is done currently. Multispeaker TTS systems like RTVC(https://github.com/CorentinJ/Real-Time-Voice-Cloning) could have been used to add more variations.

4. The authors don't identify the speakers of each of the audio files due to the non-collection of name/email id. However, speaker ID is an important feature in such a dataset. The speaker-id could have helped concretely count the number of unique speakers and could have been used for other downstream tasks. Unique information of each person could have been collected which may have been not released. However, the speaker id could have been obtained and released publicly.

5. I think the paper is not well written. Multiple sections need significant improvement. Diagrams should be used to differentiate between the three baseline methods to explain them more clearly. Section 5.1 should also be written more clearly.

6. Dataset papers should have more insights about the dataset itself. Very little or no information is provided about the demographics of the speakers, gender of the speakers, age of the speakers, vocabulary, audio characteristics like sampling rate, etc. Ideally, the authors should provide more of such information.

**Additional Feedback:**

None

**Clarity:**

I think the paper is not well written. Multiple sections need significant improvement. Diagrams should be used to differentiate between the three baseline methods to explain them more clearly. Section 5.1 should also be written more clearly. Dataset papers should have more insights about the dataset itself. Very little or no information is provided about the demographics of the speakers, gender of the speakers, age of the speakers, vocabulary, audio characteristics like sampling rate, etc. Ideally, the authors should provide more of such information. The authors also don't show a clear comparison table with the previous datasets for this problem. Even though they cite the relevant datasets, a comparison between the number of hours available in the previous dataset and the proposed one will be a good addition to the paper.

**Correctness:**

The dataset has been collected reasonably well. I find the dataset collection technique to be fine and have no complaints. According to the authors, the dataset collection could have been done in an alternative way which is better suited for real-world usage but much more complex.

**Documentation:**

The authors have clearly mentioned the collection process in the supplementary material. The data is already hosted publicly and the codebase is being used as well. The authors don't mention anything about maintaining or updating the dataset with more data in the future.

**Ethics:**

The paper is ethically fine with no major concerns.

**Relation To Prior Work:**

The prior works don't have numerical values spoken in the recorded sentences, which this dataset proposes. However, a table comparing with the previous dataset is missing and is a major weakness, in my opinion.

**Summary And Contributions:**

The work deals with creating a Spoken Language Understanding dataset. The unique feature of this dataset is that it contains numbers which previous datasets don't contain. This allows the spoken language understanding algorithms to potentially expand to basic numerical operations like unit conversion and simple mathematics. The dataset is collected using a web-based platform. The authors have benchmarked three techniques on the dataset using two metrics. The dataset and the related codebase is released by the authors publicly.

---

> ### Author Response · Authors · 2021-07-09
> **Response**
>
> Thanks for your review! We'll try to address the points you raised, and hopefully that will change your mind re: rejection.
>
> > The dataset contains only 2.5 hours of real speech (only 1.9 hrs in the train set). The authors use synthetically generated data to augment the dataset which is 68 times more than the real data. Such less amount of real speaker data is a weakness in my view for a dataset paper.
>
> Agreed, the dataset is fairly small, but as we argue in Sec 4.5 (and as shown by the results in Table 5), it is more than large enough (especially when all-real is used for testing) to support interesting experiments on a domain for which there currently exist no other resources.
>
> > The authors don't show a clear comparison table with the previous datasets for this problem. Even though they cite the relevant datasets, a comparison between the number of hours available in the previous dataset and the proposed one will be a good addition to the paper.
>
> We can add such a table, but we believe the importance of Timers and Such is not the size of the dataset (as we note, it is much smaller than some existing open-source SLU datasets), but rather the domain it covers.
>
> > The synthetic data can be ideally collected with more variations than what is done currently. Multispeaker TTS systems like RTVC(https://github.com/CorentinJ/Real-Time-Voice-Cloning) could have been used to add more variations.
>
> That's a good idea. We do use a multispeaker TTS with 22 speakers (the VoiceLoop system trained on VCTK, Sec 4.4), but we could add even more voices using the RTVC system. Thanks for the suggestion!
>
> > The authors don't identify the speakers of each of the audio files due to the non-collection of name/email id. However, speaker ID is an important feature in such a dataset. The speaker-id could have helped concretely count the number of unique speakers and could have been used for other downstream tasks. Unique information of each person could have been collected which may have been not released. However, the speaker id could have been obtained and released publicly.
>
> While we don't store any personally identifiable information, in the dataset .csv files we do list the recording session ID along with each utterance, which can be approximately treated as speaker ID. (This isn't something we mentioned in the paper; we'll add a note about it.) You're right that having an actual speaker ID would have been better for the purposes of some downstream tasks, but as we noted in Sec 4.2, we are at least guaranteed by construction that there is no speaker overlap between the train and test sets.
>
> > I think the paper is not well written. Multiple sections need significant improvement. Diagrams should be used to differentiate between the three baseline methods to explain them more clearly. Section 5.1 should also be written more clearly.
>
> We can add some diagrams like [this](https://lorenlugosch.github.io/images/slu/decoupled.png) and [this](https://lorenlugosch.github.io/images/slu/direct.png) to illustrate the method, if that helps to clarify Sec 5.2. Re: "Multiple sections need significant improvement", can you please describe more clearly what you feel is not written well, e.g. in Sec 5.1?
>
> > Dataset papers should have more insights about the dataset itself. Very little or no information is provided about the demographics of the speakers, gender of the speakers, age of the speakers, vocabulary, audio characteristics like sampling rate, etc. Ideally, the authors should provide more of such information.
>
> Can you clarify what might be missing here? We do list the speakers' English proficiency in Table 3, gender in Table 2 (except for synthetic speakers, for whom this information is not given by the original paper for the speech synthesizer), age in Table 4 (with coarse granularity to save space; the exact ages are given in *-demographics.csv), the exact script (and vocabulary) used to generate all the prompts in the gist in footnote 4, the audio characteristics (single-channel 16,000 Hz .wav) in Sec 4.3, and the speaker counts and recording statistics in Table 1.

---

> ### Author Response · Authors · 2021-07-11
> **Revision**
>
> In the revised paper, we've incorporated your feedback:
> - we added a diagram explaining the sequence-to-sequence architecture used by the three baselines (Appendix C, in the supplementary zip file)
> - suggestion for RTVC multispeaker TTS added to the future work
> - table comparing Timers and Such with existing SLU datasets (also in Appendix C)

---

> > ### Comment · Reviewer_Bag3 · 2021-07-12
> > **Response to revised paper**
> >
> > Thanks for replying to my comments in detail. I believe my concerns have been answered well to a great extent. After going through your response, I am leaning more positively about the paper.

---

> > > ### Author Response · Authors · 2021-07-13
> > > **Score**
> > >
> > > Great, glad to hear that. Given that we addressed your concerns, could we ask you to update your score to an Accept, please? Thank you!

---

### Official Review · Reviewer_WU6K · 2021-07-04
**A specific dataset for numbers with small real data and large synthetic data.**

**Rating:** 6
**Confidence:** 4
**Clarity:** This paper is well written.

**Strengths:**

1. A specific dataset for spoken English commands involving numbers.
2. Although the real data is small, but a large synthesis data is included.
3. Extensive experiment is conducted.

**Weaknesses:**

1. The real data is relatively small.
2. The text is generated with simple grammar, and may not cover different expressions.

**Additional Feedback:**

1. I think maybe instead of directly train the multistage model from scratch, maybe train the decoupled models first and then use them to initialize the multi-stage model is a good way to try.
2. The SLU output is a structured output. How to make it sequential to train a sequence to sequence SLU model is not mentioned.


**Correctness:**

I still think it is a correct way to use the method mentioned by the author to collect the data， aka, ask the volunteers what they would say to complete a certain task. If it is too random or complex, maybe a part of the data can be collected in this way.

**Documentation:**

The documentation is good.

**Relation To Prior Work:**

Yes, it is clearly discussed the difference from previous contributions.

**Summary And Contributions:**

This paper introduces a dataset of spoken English commands involving numbers, aka Timers and Such. Although the real data is small, a supplemental dataset with synthetic speech is also included. The extensive experiment is conducted on this dataset with different kinds of baseline methods, including decoupled, multistage and direct SLU models.

---

> ### Author Response · Authors · 2021-07-11
> **Response**
>
> Thanks for your review! Some responses to the issues you raise:
>
> > The real data is relatively small.
>
> Agreed, in that "test-real" is fairly small, but this is somewhat helped by using "all-real" as a large test set, similar to the setup for the smaller Snips dataset we mention (see Sec 4.5).
>
> > The text is generated with simple grammar, and may not cover different expressions. ... I still think it is a correct way to use the method mentioned by the author to collect the data， aka, ask the volunteers what they would say to complete a certain task. If it is too random or complex, maybe a part of the data can be collected in this way.
>
> Agreed. We'd like to try this in the future, but like we said, it's much more difficult to set that up.
>
> > I think maybe instead of directly train the multistage model from scratch, maybe train the decoupled models first and then use them to initialize the multi-stage model is a good way to try.
>
> That's a good idea. In fact, the way our code is written (https://github.com/speechbrain/speechbrain/blob/e585dd5faac8da60ad6372aa60ff819aaa8fd407/recipes/timers-and-such/multistage/train.py#L48), it would be very easy to try this, by accessing the "batch.transcript" field instead of the ASR output during the first few epochs.
>
> > The SLU output is a structured output. How to make it sequential to train a sequence to sequence SLU model is not mentioned.
>
> We do mention this ("the semantic dictionaries are treated as raw sequences of characters and split using a 51-token SentencePiece tokenizer"), but maybe we could make it more obvious. Something like [this diagram](https://lorenlugosch.github.io/images/slu/nlu.png)?

---

> ### Author Response · Authors · 2021-07-11
> **Revision**
>
> In the revised paper, we've incorporated your feedback:
> - re: "maybe train the decoupled models first and then use them to initialize the multi-stage model is a good way to try", we've added this suggestion to the future work, as a possible avenue for speeding up the multistage system (at train time)
> - on how the structured outputs are made sequential, we've also added a diagram showing how the characters of the label are predicted sequentially (see Appendix C in supplementary material)

---

### Official Review · Reviewer_YB1P · 2021-07-04
**Dataset for an essential processing of numbers, described with a well-organized building pipeline**

**Rating:** 7
**Confidence:** 4
**Clarity:** The paper is well written and easily …

**Strengths:**

- The proposed benchmark handles numbers in spoken language understanding, which is important for a real world application but has been underestimated
- The categorization of number information, namely SetTimer, SetAlarm, SimpleMath, and UnitConversion, reflects the needs of SLU context
- Design process is demonstrated step by step, from the initiatives to the expansion, telling the readers how the authors reached the desired amount of data with a guaranteed quality
- The collection process describes in detail the considerations conducted while recruiting and cleansing; especially encouraging the community contribution and managing the possibly noisy instances will help future researches collect the dataset with a guaranteed quality

**Weaknesses:**

- The paper lacks the literature study. It would be better if there is a survey on the struggles in SLU context that deals with temporal and numeric features, or in other view, MRC materials that handle temporal and mathematical reasoning
- Char level F1 in evaluation section may have to be more elaborately designed, but it is not necessarily the scope of this study

**Additional Feedback:**

- Regarding the char F1 evaluation, it is not intuitive that '111' and '210' have same amount of error compared to '110', given that 111 is much more close to 110 than 210 does.
- Also, given that this benchmark concentrates on numerics, it seems that some kind of normalization or digit shifting is required in the evaluation process.
- It would be nice if there is explanation that reporting SLU WER can compensate the weakness of char level evaluation.

**Correctness:**

I guess the evaluation should be more elaborately designed and described, for this study to be a benchmark. Other parts (motivation, preliminary, construction, experiments) are quite clear.

**Documentation:**

The process of benchmark construction is well documented. It is nice that the dataset is available in a recently developed speech processing toolkit.

**Ethics:**

Ethical concerns are mostly resolved by the authors' approaches and the statement.

**Relation To Prior Work:**

The paper lacks the discussion on how the proposed benchmark is related to the literature, but it does not necessarily harm the completeness of the paper.

**Summary And Contributions:**

This paper provides a dataset for spoken language understanding with numbers, `Timers and Such', and its detailed construction process. Though it lacks a literature search, its necessity is quite well demonstrated, and it is clear that this kind of benchmark has been rarely suggested so far. The paper describes the initiative attempts, how the authors extended their work to a scalable size, and which factors they had taken into account while collecting the volunteered recordings.

---

> ### Author Response · Authors · 2021-07-11
> **Response**
>
> Thanks for the review! Some responses:
>
> > The paper lacks the literature study. It would be better if there is a survey on the struggles in SLU context that deals with temporal and numeric features, or in other view, MRC materials that handle temporal and mathematical reasoning
>
> We're not aware of any published work on this specific issue! That's one reason we think this dataset will be so valuable. (But if you have any specific work in mind, please do let us know so we can incorporate it into the discussion.)
>
> Re: reasoning, we think that might not be relevant here (since the system doesn't actually have to learn to _do_ the math, it just has to figure out the arguments---see Listing 1), but we're open to discussing that type of work if you think there's another connection.
>
> > I guess the evaluation should be more elaborately designed and described, for this study to be a benchmark. Other parts (motivation, preliminary, construction, experiments) are quite clear.
>
> Thanks for the feedback. We can try to make the evaluation section a bit clearer (see below).
>
> > Regarding the char F1 evaluation, it is not intuitive that '111' and '210' have same amount of error compared to '110', given that 111 is much more close to 110 than 210 does.
>
> Maybe we didn't put this clearly in the paper: we totally agree. It doesn't really make sense to report character-level error (see the appendix responding to Interspeech reviewers), since even a small error in the numbers will totally break something like a cooking timer or a calculation. The reason we also report SLU WER is that the system could get the _numbers_ right, but the other slots/slot values wrong, so it might also be worthwhile to measure how well the system is performing in terms of all slots. We'll see if we can make that clearer.
>
> > Char level F1 in evaluation section may have to be more elaborately designed, but it is not necessarily the scope of this study
>
> (See above.)
>
> > It would be nice if there is explanation that reporting SLU WER can compensate the weakness of char level evaluation.
>
> (See above.)
>
> > Also, given that this benchmark concentrates on numerics, it seems that some kind of normalization or digit shifting is required in the evaluation process.
>
> Numbers, and everything else, are represented as sequences of raw characters (12.5 = "1,2,.,5"), so there's really only one way the model can learn to output a number, unlike with words, where there might need to be some kind of normalization ("twelve and a half", "twelve point five"). (Another reviewer was unclear on this, so we can try to make that fact more obvious.)

---

> ### Author Response · Authors · 2021-07-11
> **Revision**
>
> In the revised paper, we've incorporated your feedback:
> - the evaluation section has been made clearer, with a note about character-level evaluation
> - note on how normalization of numerics is handled automatically

---

> ### Comment · Reviewer_YB1P · 2021-07-20
> **Checked the revision**
>
> I've checked the updates in the revised paper :)

---

### Official Review · Reviewer_whss · 2021-07-04
**An important SLU dataset dealing with multi-digit numbers in realistic contexts**

**Rating:** 7
**Confidence:** 3

**Strengths:**

1. Coverage of multi-digit scenarios: Although there is no dearth of such datasets in NLP as they are often used to test models vis-a-vis common sense reasoning and related tasks, the lack of them in the spoken form is an important gap that needs to be filled, in my opinion. Understanding voice commands can often be successfully dealt in two stages by first transcribing the voice and then understanding the transcript, as shown by the multistage baseline, but for that, one first needs a dataset of spoken voice commands from which the transcript can be generated. This paper tries to fill that gap by creating a dataset of real and synthetic voices that cover various scenarios in daily human lives where people deal with multi-digit numbers.

2. Benchmarking baselines: the paper extensively evaluates different important baselines on the dataset on both the real and synthetic versions of the dataset. This gives insights into the relative usefulness of different baseline types and also shows that authors have done a decent job even with the real data collection despite its limited amount in comparison to the synthetic data.

**Weaknesses:**

Major:
1. As the authors point out, the synthetic version of the dataset is quite artificial in that the speeches are from experts and recorded in very controlled scenarios. However, in the real world it might be very hard to find such controlled environments. If that's the case, why didn't the authors try to collect more real-world data through better forms of crowdsourcing like mechanical Turks, paid volunteers, etc. Even if the models trained mainly on synthetic data generalize very well to the real-world data, the real-world test set should be larger than just 10 speakers and 0.3 hours of audio. Some discussion on this aspect would surely help the paper's cause.

Minor:
1. Page 4, 4.2 Speaker Recruitment: I am not sure what issues there might be with collecting speaker information that helps to identify them if the authors of the dataset keep them anonymous during the dataset release. Like the authors pointed out, that would help them remove the duplicates and make the dataset even cleaner. Besides, asking a volunteer to submit their voice samples only once is a completely legit request.

2, Need for decoupled baseline: in which scenario would one require the decoupled model? If I understand correctly, the only difference between the decoupled model and the multistage model with precomputed transcript is that in the decoupled model, the NLU part of the model has access only to the ground truth transcripts during training, which significantly worsens its performance. However, I can't think of any case where the NLU model won't have access to the ASR outputs during training. Even in privacy preserving cases where distributed or federated learning needs to be used, the NLU model can very well be expected to have access to the ASR outputs during training as they absolutely need those during inference.


P.S.: I am leaning towards acceptance but would like to see the above-listed concerns addressed.

**Additional Feedback:**

None

**Clarity:**

The paper is well-written on the whole.

However, some clarifications could improve the paper:
1. Page 4, 4.4 Synthetic Data: the paper talks about 'redundant rows' being part of the .csv file for the sake of balancing. Which .csv file is this (don't think it has been described before)? Also, if the .csv is part of the opensourced dataset, I don't see why one would maintain copies of the same data sample in that. One can always have unique rows in the csv and do the balancing during the data-loading. That allows for different flexibilities: balancing on the fly, weighing the losses instead of balancing the data, etc.

2. Page 7, 5.4, "can overfit to synthetic speech" in the title: why use the term 'overfit' when the direct model also performs better when trained on 'train-real' and tested on 'test-real'?

**Correctness:**

The claims made in the paper and the supporting experiments/evaluation methods look correct to me.



**Documentation:**

The dataset looks well-documented.

**Ethics:**

I couldn't spot any ethical issues.

**Relation To Prior Work:**

The work extensively compares with prior work and tries to address their shortcomings.

**Summary And Contributions:**

This work presents a dataset for SLU that focuses on voice commands dealing with multi-digit numbers, which often find use in the daily activities of humans. The proposed dataset tries to fill an important gap in current SLU benchmarks that, at best, deal with single-digit scenarios. SLU models are also important in developing and deploying home voice assistants but understanding multi-digit commands often needs access to an internet search. However, having such a dataset should allow the training of models on-the-edge that could try and understand multi-digit voice commands directly. Further, the paper extensively benchmarks different types of relevant baselines on this dataset and provides a hassle-free way to download and use the dataset.

Edit: score updated on July 13th

---

> ### Author Response · Authors · 2021-07-09
> **Response**
>
> Thanks for your review, and we're glad to see that you agree that the dataset is important and useful!
>
> > As the authors point out, the synthetic version of the dataset is quite artificial in that the speeches are from experts and recorded in very controlled scenarios. However, in the real world it might be very hard to find such controlled environments. If that's the case, why didn't the authors try to collect more real-world data through better forms of crowdsourcing like mechanical Turks, paid volunteers, etc.
>
> The synthetic dataset is actually fairly realistic, as it uses a speech synthesizer trained on the VCTK dataset, which has casual/everyday speech (not actors/experts, like services such as Google's Cloud TTS). For your convenience, I've uploaded a couple samples [here](https://drive.google.com/drive/folders/1iUMyt5SrWD4jXbv_mDDEwtlE8UO55k_T?usp=sharing) so you can judge for yourself.
>
> Re: MTurk/paid speakers, we just didn't have any budget to set that up. (We did reach out to Scale AI when that company put out a call offering to fund the creation of open-source datasets, but they informed us that their funding was specifically focused on image datasets.) It's something we'd consider doing for Timers and Such v2.0, if we got our hands on enough money for it.
>
> > Even if the models trained mainly on synthetic data generalize very well to the real-world data, the real-world test set should be larger than just 10 speakers and 0.3 hours of audio.
>
> Agreed. This is the purpose of the "all-real" subset for testing: it's a lot larger (all 95 speakers/2,151 utterances/2.5 hours) and should give more reliable performance estimates.
>
> > Page 4, 4.2 Speaker Recruitment: I am not sure what issues there might be with collecting speaker information that helps to identify them if the authors of the dataset keep them anonymous during the dataset release. Like the authors pointed out, that would help them remove the duplicates and make the dataset even cleaner. Besides, asking a volunteer to submit their voice samples only once is a completely legit request.
>
> One issue is that such information needs to be stored securely and care needs to be taken that it doesn't get released; the advantage of not storing personally identifiable information is that maintaining the dataset is simpler, and more speakers participate who might otherwise be hesitant to. But you're absolutely right: as we wrote in Sec. 4.2, we could have asked speakers to record only one session; it just didn't occur to us to do so.
>
> > Need for decoupled baseline: in which scenario would one require the decoupled model? ...
>
> Very often, you only have text with NLU labels, and no speech data (or rather no speech data specifically labeled with the desired intents). You can get speech data to train an end-to-end model by just using a speech synthesizer on that text, but 1) you might not have a good speech synthesizer for your language, and 2) it is significantly more expensive to train on speech than on text (as we note in Sec. 5.2). So it's still worthwhile to think about decoupled models and how to improve their performance, which is why we include that baseline. (These numbers also serve to show for the first time, on open-source data, that the multistage approach works better than the decoupled approach. This had only ever been shown for Google's closed-source data in [28].)
>
> > Page 4, 4.4 Synthetic Data: the paper talks about 'redundant rows' being part of the .csv file for the sake of balancing. Which .csv file is this (don't think it has been described before)?
>
> This is the .csv files containing the audio filenames, transcripts, and semantic labels. Thanks for pointing this mistake out; we'll be sure to add that to the paper.
>
> > Also, if the .csv is part of the opensourced dataset, I don't see why one would maintain copies of the same data sample in that. One can always have unique rows in the csv and do the balancing during the data-loading. That allows for different flexibilities: balancing on the fly, weighing the losses instead of balancing the data, etc.
>
> You're right; the only issue is the user would have to identify these rows and balance them manually---the way we've left it, the dataset is already perfectly balanced.
>
> > Page 7, 5.4, "can overfit to synthetic speech" in the title: why use the term 'overfit' when the direct model also performs better when trained on 'train-real' and tested on 'test-real'?
>
> We wanted to emphasize that this model gets nearly perfect results on test-synth not because it is suddenly a much better model, but rather because it has just memorized what the synthesizer sounds like. Agree that "overfit" is probably a misnomer here, since this model still performs well on test-real; we'll think of a better way to express this.
>
> > P.S.: I am leaning towards acceptance but would like to see the above-listed concerns addressed.
>
> Thanks, and hopefully we've addressed all your concerns!

---

> > ### Comment · Reviewer_whss · 2021-07-13
> > **Reply to responses**
> >
> > I believe my questions have been adequately addressed and I am willing to increase the score during the discussion period.

---

> ### Author Response · Authors · 2021-07-11
> **Revision**
>
> In the revised paper, we've incorporated your feedback:
> - the .csv files now have a description
> - we've rephrased the misleading "overfitting" statement
> - we added a note describing how a user might "un-balance" the data by removing the redundant rows, if desired

---

> ### Author Response · Authors · 2021-07-13
> **Score**
>
> If you feel that we addressed your concerns with our response and updated paper, could we ask you to increase your score, please? Thank you!

---

> > ### Comment · Reviewer_whss · 2021-07-13
> > **Re: increase scores**
> >
> > Yes, I will do that.

---

### Author Response · Authors · 2021-07-11
**Some points to remember**

Thanks to all the reviewers for taking the time to read and review the Timers and Such paper.

In addition to the responses we wrote to the reviews, we'd like to emphasize a few points:

- At 2,151 utterances, the real dataset is small compared to datasets like LibriSpeech. But small datasets can still be very useful for research and development. In fact, the usual scenario in SLU is that there is relatively little semantically labeled audio available, but a large amount of general-purpose audio and semantically labeled text. Training on the 192,000 synthetic utterances and testing on the 2,151 real utterances (Table 6) mirrors this scenario. Still, the results in Table 5 show that the real subset is already large enough to train models that perform well.

- While the real subset of Timers and Such is small in terms of the number of utterances, it has a very demographically diverse set of 95 speakers (see Tables 2,3,4). This diversity is lacking in some commonly used speech benchmarks, and we think it's important to encourage the publication of datasets like Timers and Such that do feature non-native speakers using everyday devices, as opposed to professional recording setups.

- Our contribution is not only the data: it's also the extensive code and pre-trained models for the baseline experiments. The code enables easy comparison between the conventional decoupled SLU setup and end-to-end SLU setups, for which there has been no open-source code in the past. And the code can be used not only for Timers and Such: users can apply it to _any_ SLU dataset for which the output labels take the form of a dictionary, since the code converts this dictionary (no matter what format it has) into a sequence of characters that gets predicted by the seq2seq model. (In fact, this is how we created the state-of-the-art SpeechBrain recipes for the SLURP and Fluent Speech Commands SLU datasets.)

We hope the reviewers and organizers will keep these points in mind while making a decision---we're excited by the prospect of the Datasets and Benchmarks track, and hope our paper will appear in it!

---

### Author Response · Authors · 2021-07-11
**Revision uploaded**

We've uploaded a new version of the paper, with changes/fixes suggested by the reviewers in blue (including a new appendix in the supplementary .zip file).

---

### Author Response · Authors · 2021-07-13
**Updating scores**

Thanks to Reviewer Bag3 for updating their score to an Accept based on our response. If any of the other reviewers felt that their opinion of the paper was improved by our responses and revised paper, we'd ask them to please update their scores, as well. Thank you!

---

### Decision · Program_Chairs · 2021-07-26

**Decision:**

Accept

**Comment:**

This paper introduces a speech dataset that involves multi-digit numerics. Particularly, they are looking into four scenarios (set timer, unit conversion, set alarm, simple math operations) that involve multi-digit utterances. Along with a small amount of (2.5 hours of human speech) real speech data, the authors also present large-scale synthetic data. Then, the authors present extensive experiments with decoupled, multistage, and direct SLU models. The dataset fills in the gap of what's missing in current datasets -- which does not cover such numeric expressions. The reviewers all agreed that this can provide useful resources to study the understudied, yet important problem of numeric expression understanding.

The data collection method is fairly standard, i.e., speaking out what's written, instead of simulating situations and acquiring free speeches. From a language perspective, it's fairly simple -- as it is generated from four simple scenarios/grammar. This is reasonable as the task is focused on numeric understanding, and it's not easy to find such natural texts without access to commercial platforms. Yet, this is somewhat disappointing, and for future revisions of the data, it'll be worthwhile to study a two-stage approach where the user can paraphrase the generated sentence. The reviewers also have concerns about the size of the dataset, but for evaluation purposes, this could be sufficient.